# Melatonin Improves Levels of Zn and Cu in the Muscle of Diabetic Obese Rats

**DOI:** 10.3390/pharmaceutics13101535

**Published:** 2021-09-22

**Authors:** Miguel Navarro-Alarcón, Fernando Gil-Hernández, Cristina Sánchez-González, Juan Llopis, Marina Villalón-Mir, Pablo Olmedo, Pablo Alarcón-Guijo, Diego Salagre, Lorena Gaona, Mario Paredes, Ahmad Agil

**Affiliations:** 1Department of Nutrition and Bromatology, School of Pharmacy, University of Granada, 18016 Granada, Spain; marinavi@ugr.es; 2Department of Toxicology, School of Medicine, University of Granada, 18071 Granada, Spain; fgil@ugr.es (F.G.-H.); polmedopalma@ugr.es (P.O.); 3Department of Physiology, Institute of Nutrition and Food Science, Biomedical Research Centre, University of Granada, 18071 Granada, Spain; crissg@ugr.es (C.S.-G.); jllopis@ugr.es (J.L.); 4Department of Pharmacology, Biohealth Institute and Neurosciences Institute, School of Medicine, University of Granada, 18016 Granada, Spain; palarcon@ugr.es (P.A.-G.); dsalagre@ugr.es (D.S.); lgaona1972@hotmail.es (L.G.); md.marioparedes@gmail.com (M.P.); 5Biohealth Research Institute in Granada (ibs. GRANADA), University Hospital of Granada, 18016 Granada, Spain; 6Department of Teaching of the Nursing Career of the Catholic University of Santiago de Guayaquil, General Teaching Coordination Hospital of Specialties Dr. Teodoro Maldonado Carbo of Ecuador, Guayaquil 090615, Ecuador; 7Department of Docent, Faculty of Medicine, Catholic University of Santiago de Guayaquil, Epidemiology and Tropical Medicine Services, Directorate of Naval Health, Navy of Ecuador, Guayaquil 090615, Ecuador

**Keywords:** melatonin, zinc, copper, diabetic obese rat, muscle and other organ and white adipose tissues

## Abstract

Melatonin improves metabolic alterations associated with obesity and its diabetes (diabesity). We intend to determine whether this improvement is exerted by changing Zn and/or Cu tissue levels in liver, muscle, pancreas, and brain, and in internal (perirenal, perigonadal, and omentum) and subcutaneous lumbar white adipose tissues (IWAT and SWAT, respectively). Male Zücker diabetic fatty (ZDF) rats and lean littermates (ZL) were orally supplemented either with melatonin (10 mg/kg body weight/day) or vehicle for 6 weeks. Zn and Cu concentrations were not significantly influenced by diabesity in the analyzed tissues (*p* > 0.05), with the exception of Zn in liver. In skeletal muscle Zn and Cu, and in perirenal WAT, only Zn levels increased significantly with melatonin supplementation in ZDF rats (*p* < 0.05). This cytoplasmic Zn enhancement would be probably associated with the upregulation of several Zn influx membrane transporters (Zips) and could explain the amelioration in the glycaemia and insulinaemia by upregulating the Akt and downregulating the inhibitor PTP1B, in obese and diabetic conditions. Enhanced Zn and Cu levels in muscle cells could be related to the reported antioxidant melatonin activity exerted by increasing the Zn, Cu-SOD, and extracellular Cu-SOD activity. In conclusion, melatonin, by increasing the muscle levels of Zn and Cu, joined with our previously reported findings improves glycaemia, insulinaemia, and oxidative stress in this diabesity animal model.

## 1. Introduction

Melatonin is a potent antioxidant, and its presence in numerous plants and foods has been described as beneficial to human health [1]. Melatonin is also a hormone synthesized in multiple organs such as the pineal gland, which functions during the night period and regulates the circadian rhythm [2,3]. It has been indicated that the decrease of endogenous melatonin levels is associated with the increased risk of obesity, and the associated diabetes mellitus [4,5,6,7,8]. Moreover, it has been reported that Zn has an important role in the endogenous melatonin synthesis in the pineal gland [9]. On the other hand, the increase of visceral white adipose tissue produces insulin resistance which affects skeletal muscle and liver metabolism [10,11,12,13], eventually producing chronic hyperglycemia [3].

In several studies, Zn has been described to act insulin-likely in skeletal muscle cells. It has been found that this activity is associated with the families of the different Zn transporters [14,15], which are responsible for its traffic through biological membranes [16], as well as metallothioneins that link it at the intracellular level [14]. These researchers [15] reported that Zn administration stimulated glucose oxidation and glycemic control by modulating the insulin signaling pathway in human and mouse skeletal muscle cell lines. Similarly, Lu et al. [17] demonstrated that protamine Zn insulin combined with sodium selenite improved glycometabolism in T2DM via upregulating phosphatidylinositol 3-kinase (PI3K), downregulating the protein tyrosine phosphatase 1B (PTP1B: Inhibitor of the insulin receptor substrates 1 and 2 [Irs1 and Irs2]), reducing oxidative stress, and ameliorating skeletal muscle and β-cell damage, as well as mitochondrial dynamics.

In relation to Cu, it has been reported that disturbances in their levels in several organs and biofluids are correlated with abnormalities implicated in metabolic pathways of diabetes and its complications [18,19]. Zhou et al. [20] reported that in a type 1 diabetic mice model induced by streptozotozin, the renal and serum Cu levels were significantly higher, while the hepatic Cu and Zn levels were significantly decreased at 2 weeks and 2 months, respectively, after the diabetes onset.

On the other hand, in T2DM, there is a marked meta-inflammation and oxidative stress, which has been related to decreased expression and activity of cytoplasmic (Zn, Cu-SOD: SOD1), mitochondrial (Mn-SOD: SOD2), and extracellular (Cu-SOD: SOD3) superoxide dismutase (SOD) enzyme, an effect observed in different tissues such as liver, skeletal muscle or kidney of diabetic animal models [17,21,22,23,24]. Among the multiple pathways through which melatonin could exert its antioxidant effect, as described by Zhan and Zhang [25], the activation of the SOD1 and SOD2 antioxidant enzymes is indicated, or even, the reactivation of these enzymes in different animals as well as pathological conditions [25,26,27].

In multiple studies on melatonin supplementation, this indolamine has been reported to limit obesity, improve chronic hyperglycemia, and reverse the problems of glucose intolerance, insulin resistance, oxidative stress, and inflammation [3,10,26,27,28,29,30,31,32,33]. However, whether the positive effect that melatonin supplementation has been shown to exert on blood glucose levels [3], insulinaemia [34], and oxidative stress [26] under conditions of obesity and associated diabetes, is established through an intermediate signaling pathway mediated by Zn and/or Cu levels in several tissues such as liver, muscle, pancreas, brain or WAT, has not been done so far. Therefore, the aim of the present work is to study the possible influence of melatonin supplementation on Zn and Cu levels in liver, muscle, pancreas, and brain, and in internal (perirenal, perigonadal, and omentum) and subcutaneous lumbar white adipose tissues (IWAT and SWAT, respectively) of an animal model of obesity and associated T2DM (ZDF rats). Therefore, our study provides new insight into the possible signaling pathways through which melatonin supplementation could exert the beneficial effects described [35] in conditions of obesity and diabetes [2].

## 2. Experimental Section

### 2.1. Reagents

Melatonin was obtained from Sigma Chemicals Madrid, Spain. Commercially available standard solutions of Zn and Cu (1000 mg/L; Tritisol, Merck, Darmstadt, Germany) were used to prepare calibration graphs. All of the solutions were prepared from analytical grade reagents (Suprapur, Merck): 65% HNO_3_, 65% HClO_4_. Double-distilled deionized water with a specific resistivity of 18 mΩ/cm was used to prepare standards for calibration and dilutions, and was obtained immediately before use by filtering distilled water through a Milli-Q purifier (Millipore, Waters, Mildford, MA, USA).

### 2.2. Animals and Experimental Protocols

The experiment was approved by the Ethical Committee of the University of Granada (Granada, Spain) according to the European Union guidelines. The permit project number is 409-2016-CEEA.

The experiment was performed on 40 Zucker male rats: 20 ZDF rats (a 2-type DM model) weighing 180–200 g, and 20 ZL rats weighing 120–140 g. They were bought at 5 weeks old from Charles River Laboratories (Barcelona). The rats were fed with Purina chow whose composition was: Carbohydrates (58.5%), fat (6.5%), proteins (23%), fiber (4%), and ash (6.8%). Animals had the ready diet and tap water with “*ad libitum*” access [36]. The diet was administrated for 6 weeks, and the proportion of food intake by groups ZDF/ZL was approximately 2.5 times [37]. The animals were housed 3–4 rats/plastic cage and kept under controlled 12 h light-dark cycle (lights were switched on at 07:00 h).

At the age of 6 weeks, the animals were randomly divided into four groups (*n* = 10 per group): The non–supplemented control groups (C–ZDF and C–ZL) and the melatonin-supplemented groups (M–ZDF and M–ZL). Melatonin was dissolved in a minimum volume of absolute ethanol. Then, it was diluted to the final solution of 0.066% (*w*/*v*) in the drinking water to yield a dose of 10 mg/kg body weight (bw) and was received daily for 6 weeks. The animals in the non–supplemented control groups received the vehicle in the drinking water at a comparable dose and supplementation duration. Fresh melatonin and vehicle solutions were prepared every 2 days. Moreover, the melatonin dose was adjusted for body weight over the entire period of the study [37], taking into consideration that every 2 days the rats were weighed again and the corresponding adjustments were made. Water bottles were covered with aluminum foil to protect from light, and the drinking fluid was changed twice weekly.

At the end of the experiment, the animals were sacrificed by intraperitoneal administration of sodium phenobarbital (thiopental) anesthesia at 40 mg/kg dose. The liver, muscle, pancreas, and brain were collected by surgical excision. Internal WAT fat pads (omental, perigonadal, and perirenal fat) and lumbar subcutaneous fat were visually inspected and also collected by surgical excision. All of the tissues were washed thoroughly in a saline solution and stored at −80 °C for further experiments.

### 2.3. Zn and Cu Determination by Flame Atomic Absorption Spectrometry (AAS)

A Perkin-Elmer 1110B double-beam atomic absorption spectrophotometer equipped with a Zn or with a multi-element Cu hollow cathode lamp were used together (Perkin-Elmer, Norwalk CT, USA). A thermostatic multiplace digestion block (Selecta, S.A., Barcelona, Spain) was also employed.

The Zn and Cu levels in the studied tissues (liver, muscle, pancreas, brain, and perigonadal, perirenal, omental, and lumbar subcutaneous fat) were determined by weighing 0.100–0.200 g of the samples in Pyrex glass tubes, adding 0.8 mL HNO_3_, heating at 80 °C for 15 min, and then at 180 °C for 45 min. Next, 0.8 mL of an HNO_3_–HClO_4_ solution (4:1) was added and they were heated at 200 °C for 90 min. The resulting digested solution was then diluted to 2.5 mL with Milli-Q water and maintained in this state until analysis [38].

The Zn and Cu levels were determined by flame (direct aspiration) atomic absorption spectrometry. In an accuracy test, the Zn and Cu concentrations measured by this method in the certified reference material (Contox Trace Serum Metal Control-A Level I, Kaulson Laboratories Inc., NJ, USA [78.8 ± 4.3 and 91.8 ± 1.4 µg/dL], respectively) did not significantly differ (*p* > 0.05) from the certified levels [80.0 ± 6.0 and 90.0 ± 7.5 µg/dL, respectively]) [36].

### 2.4. Statistical Analysis

A statistical evaluation of the results was carried out using the computer software Statistical Package for the Social Sciences (SPSS 23.0, Chicago, IL, USA). The results have been expressed as arithmetic means ± standard error of the mean (SEM). The variance analysis and post hoc Tukey test were used for the comparison among groups. The *p* < 0.05 was statistically significant and levels of significance were labeled on the figures as follows: * *p* < 0.05; ** *p* < 0.01; *** *p* < 0.001.

## 3. Results

Figure 1 and Figure 2 exhibit the Zn concentrations, found in liver (Figure 1A), skeletal muscle (Figure 1B), pancreas (Figure 1C), and brain (Figure 1D) samples, and in IWAT (perirenal, perigonadal, and omentum adipose tissues, Figure 2A–C, respectively) and SWAT (Figure 2D), from Zucker diabetic fatty (ZDF) and their lean littermates (ZL) rats. In the present study, we have measured Zn concentrations in tissues reported for both rat groups (ZL and ZDF) after previous melatonin supplementation (melatonin groups: M–ZL and M–ZDF) or without melatonin supplementation (control groups: C–ZL and C–ZDF). The Zn concentrations did not significantly differ between C–ZL and C–ZDF rats for most of the several studied tissues, indicating that element concentrations in them were not significantly influenced by the obesity and T2DM pathologies. The only one exception was found for liver Zn concentrations, which were significantly increased in C–ZDF when compared with the C–ZL group (*p* = 0.041).

For Zn, in ZDF rats, the supplementation with melatonin significantly increased the concentrations in skeletal muscle (C–ZDF vs. M–ZDF; Figure 1B; *p* < 0.05) and perirenal WAT (Figure 2B; *p* < 0.01). Additionally, in ZL rats, melatonin supplementation significantly enhanced Zn concentrations in the liver (C–ZL vs. M–ZL; Figure 1A; *p* < 0.001).

For perirenal WAT (Figure 2B) and brain (Figure 1D) Zn levels measured in ZDF rats treated with melatonin were significantly higher than those measured in both ZL rat groups (M–ZDF vs. C–ZL group, *p* < 0.05 for both organs; M–ZDF vs. M–ZL group, *p* < 0.001 and *p* < 0.01, respectively). Moreover, Zn concentrations found in omentum WAT (Figure 2C) for ZDF rats treated with melatonin were significantly higher than those measured for ZL rats also treated with melatonin (M–ZL vs. M–ZDF group; *p* < 0.05; Figure 2C). Another striking finding is that from the eight analyzed tissues, for five (liver, pancreas, brain, and perirenal and subcutaneous lumbar WAT), significantly higher Zn concentrations were measured in the melatonin-treated ZDF rats when compared with those found in non-treated ZL rats (M–ZDF vs. C–ZL; *p* < 0.001 for liver and *p* < 0.05 for the remaining reported tissues). Moreover, Zn concentrations found in the brain (Figure 1D), and perirenal and omentum WAT (Figure 2B,C, respectively) for melatonin-treated ZDF were significantly higher than those found for melatonin-treated ZL rats (M–ZL vs. M–ZDF group; *p* < 0.01, *p* < 0.001, and *p* < 0.05, respectively).

Figure 3 and Figure 4 depict the Cu concentrations, respectively, measured in the same organs (Figure 3A–D, respectively) and WAT (Figure 4A–D, respectively) from the same animal groups. The Cu concentrations did not significantly differ between C–ZL and C–ZDF rats for any of the several studied tissues, indicating that element concentrations in them are not significantly influenced by the obesity and T2DM pathologies.

For Cu, in ZDF rats, the supplementation with melatonin significantly increased concentrations only in the muscle (C–ZDF vs. M–ZDF; Figure 3B; *p* < 0.05). In addition, in the liver for both groups of ZDF rats significantly higher Cu concentrations than those measured in melatonin-treated ZL rats were found (M–ZL vs. C–ZDF and M–ZDF; Figure 3A; *p* < 0.001 and *p* < 0.01, respectively). Moreover, in the liver Cu concentrations measured in melatonin, the treated ZL rats were significantly lower to those found in the non-treated ZL group (M–ZL vs. C–ZL; *p* < 0.05).

## 4. Discussion

In the present study, we found that Zn and Cu concentrations in the analyzed tissues were not affected by obesity and T2DM. This finding indicates the lack of influence of the aforementioned pathologies on the tissue homeostasis of these minerals in the animal model of diabesity ZDF rats. Similarly, other authors [39] did not find differences on Zn levels measured in the liver of ZL and ZDF groups. However, they reported that Zücker obese rats had a reduction in the Zn and Cu concentrations presumably as a result of the increased fat content in the liver of these obese rats.

Melatonin supplementation did not change the Cu levels in several peripheral organs as well as the internal and lumbar subcutaneous WAT studied, in contrast to that reported by other authors [19,40,41]. In the same way, our results did not agree with those reported by Zhou et al. [20] on the enhancement of renal and serum Cu levels and the decrease of hepatic Cu and Zn levels at 2 weeks and 2 months, respectively, after the diabetes onset, as it is induced by the streptozotozin T1DM phenotype.

In the present study, we have shown that melatonin supplementation in the diabesity animal model studied, increased Zn levels only in skeletal muscle (*p* < 0.05), as shown by comparing levels between the control group (C–ZDF) and those of the melatonin-treated group (M–ZDF).

It has been reported that Zn supplementation increased insulin receptor tyrosine phosphorylation, Akt phosphorylation, and GLUT4 translocation in skeletal muscle cells [16,42,43], which subsequently increases the uptake of glucose [10,44]. Our findings are encouraging, in the search for new classical therapeutic alternatives, given that melatonin supplementation specifically increases Zn levels in skeletal muscle, which given its insulin-mimetic character [15,45], would allow acting in the regulation of blood glucose levels in this pathology, through the activation of other Zn-dependent signaling pathways in skeletal muscle. It is also known that 60% of the total body Zn pool is in skeletal muscle [44]. Future studies should investigate whether the increase in skeletal muscle Zn following melatonin supplementation in ZDF rats is specifically established in the cytoplasm with the intervention of multiple Zn Zip transporters (1–6, 8, 10, and 14) [14,15] located in the cell plasma membrane and/or in the cytoplasmic organelles where Zip7 is located [14]. Therefore, future research should implement studies of the subcellular and compartmental levels of Zn in skeletal muscle following melatonin administration in ZDF rats. In this sense, some authors have described that overexpression of Zip7 in C2C12 cells induced multiple genes related to glucose metabolism in skeletal muscle cells, including GLUT4, Irs1 (insulin receptor substrate-1), and Irs2 [16], acting in an insulin-like manner and ultimately activating glucose uptake [46].

Therefore, future studies should determine the possible connection between melatonin supplementation, increased Zn levels in skeletal muscle, and the possible upregulation of the downregulated Zip-Zn transporters in this animal model of obesity and T2DM.

Previous studies performed by our research group have shown that melatonin supplementation (10 mg/kg/day) in the same strain model of diabesity (ZDF rats) improved blood parameters related to glucose tolerance and insulin resistance, such as fasting blood glucose, insulin, insulin resistance index such as HOMA-IR, HOMA1-%B of the β-cell function and adiponectin levels, as well as leptin concentrations [10]. One of the possible signaling pathways through which melatonin could exert such an effect in the ZDF rats would be that of increased Zn levels in skeletal muscle found in the present study, with the possible involvement of the cell plasma membrane Zip transporters (1–6, 8, 10, and 14) and/or Zip7 endoplasmic reticulum. In line with this, the insulin-mimetic action of Zn is known to act at two levels such as inactivation of the negative regulator of insulin signaling, protein tyrosine phosphatase (PTP1B) through direct binding of this enzyme [47] and that of protein kinase B (Akt) phosphorylation, which ultimately facilitates the mobilization of the glucose transporter GLUT4 to the plasma membrane with subsequent translocation of glucose into the cytosol [14,15]. In this sense, 20 µM ZnSO_4_ supplementation induced glucose oxidation in both mouse and human skeletal muscle cells compared with the untreated control [15]. The current study demonstrates a reinforcement of the aforementioned decrease in insulin resistance [48] through another additional signaling pathway such as elevated muscle Zn levels, implying finally the possible existence of two complementary pathways in the improvement of hyperglycaemia.

Other studies have reported that melatonin supplementation stimulated SOD1 in coronary artery disease [49,50] or upregulated the SOD1 to deal with the neuronal damage caused by enhanced oxidative stress induced by PCBs in adult rats [51]. Other researchers [52] stated that in patients with unstable angina, there are diminished serum levels of 6-sulfatoxy melatonin and SOD1 associated with an increase of oxidized-LDL levels as a first step to the atherosclerotic process. In addition, we have hypothesized that SOD1 activation may be mediated by the rise in muscle Zn and Cu levels observed in this study following melatonin administration in ZDF rats. Moreover, in human T2DM patients, a low expression of antioxidant enzymes such as SOD1 has been observed given the excessive glycation that sets in [53]. Extracellular SOD3 expression has also been shown to be reduced with the increased production of the superoxide radical (O_2_-•) associated with the decreased expression of the Cu transporter protein ATP7A, in blood vessels from mice with T1DM and T2DM [54], as well as in patients and those from high-fat-diet-induced T2DM [55]. In the present study, the increased Zn and Cu levels established by melatonin supplementation in muscle in ZDF rats could also be related to the increased Zn, Cu-SOD activity observed after melatonin supplementation in previous studies [17,23,24]. In this regard, melatonin supplementation in these ZDF rats has been shown to lower basal and Fe^2+^/H_2_O_2_-induced plasma lipid peroxidation by improving low-grade inflammation and oxidative stress in this animal model of obesity and T2DM [31], which could be related to an increase in Zn, Cu-SOD activity in skeletal muscle, which is primarily responsible for the uptake and expenditure of most blood glucose.

Other researchers reported that melatonin administered in vivo increased the expression of multiple oxidative stress regulatory enzyme genes in neuronal tissue, such as SOD1 and SOD2 [50,56], as well as SOD and glutathione peroxidase (GSH-Px) [57]. Additionally, Mauriz et al. [58] have pointed out a potential beneficial effect of melatonin supplementation in terms of slowing aging, as they have found that this indolamine supplementation attenuated in rat liver the downregulation in gene expression of SOD1 and cytoplasmic and mitochondrial glutathione peroxidase that occurs with aging. Despite this, and taking into account the role of Zn and Cu as enzyme cofactors of the SOD1 enzyme, in the present study we did not find that melatonin supplementation in ZDF rats modified brain and liver levels of either Zn or Cu (*p* > 0.05).

In the present study, melatonin supplementation in ZDF rats did not change the pancreatic Zn levels. For this reason, as indicated above, the improvement in T2DM in ZDF rats may have been due to the pathway previously described, which facilitated the increase in muscle glucose uptake by activation of Akt kinase phosphorylation, and the decrease in insulin resistance, as the enhancement in muscle Zn inhibited the negative regulator of the insulin receptor substrate, PTP1B, in accordance with what was established by other authors [14,15].

Of all the internal and lumbar subcutaneous WAT studied, only in the perirenal one melatonin supplementation increased Zn levels in this diabesity animal model with respect to the non-supplemented group (M–ZDF vs. C–ZDF; *p* < 0.01). Moreover, in the same animal model, we previously found that melatonin increased Ca levels in the perirenal WAT, etc. [50]. We do not know why this increase exclusively occurred in the perirenal WAT and did not in the remaining internal and lumbar subcutaneous WAT. Therefore, future studies are needed to find out which mechanism may be involved in this specific effect of melatonin. Whether a dormant BAT exists in the perirenal adipose tissue of ZDF rats, and if so, whether melatonin supplementation influences the activity of the SOD1 enzyme, one of whose enzyme cofactors is Zn, should also be studied in the future. In this diabesity animal model, previous studies already showed that melatonin supplementation increased the UCP1 protein expression in beige inguinal adipose tissues [59,60]. However, in the present study, melatonin supplementation did not change the Cu levels in perirenal WAT, a mineral that, similar to Zn, is also a cofactor of the SOD1 enzyme. On the other hand, future studies should investigate whether the increase in Zn levels after melatonin supplementation exclusively in the perirenal WAT in this diabesity animal model could be related to a higher expression of ZFP407 or even zinc-2-glycoprotein (ZAG) in the adipocytes of this WAT, whose expression is clearly reduced in adipose tissues, mainly in the subcutaneous tissue, in the pathologies of obesity [61].

A striking finding of this research is that Zn levels were increased in the melatonin-treated ZDF group with respect to the non-treated ZL one (M–ZDF vs. C–ZL), in three of the four organs analyzed (liver, pancreas, and brain), as well as in two of the four WAT studied namely perirenal and subcutaneous lumbar WAT. We hypothesized that with melatonin administration in ZDF rats (M–ZDF), and in comparison with control ZL ones (C–ZL), the oxidative stress associated with obesity and T2DM in the former was clearly increased [40]. This effect was not established in the non-melatonin-treated control ZL group. Therefore, we hypothesize that probably there was an increase in tissue Zn levels in M–ZDF, as a compensatory mechanism for this oxidative stress with a consequent increase in the expression of the cytoplasmic SOD1 enzyme. However, in future studies, it would be necessary to compare the activity and level of Zn, Cu-SOD enzyme in the studied tissues (liver, pancreas, brain, and in the perirenal and subcutaneous lumbar WAT) of the two groups of animals referred to (C–ZL and M–ZDF), in order to verify the aforementioned hypothesis.

In the present study, we have observed that melatonin administration only increased Cu levels in muscle tissue in ZDF rats. On the other hand, it is known that ATP7A as a transporter for extracellular SOD3 in vascular smooth muscle, is markedly downregulated in vessels from T2DM patients as well as those from fat-diet-induced or db/db T2DM [55] or T1DM mice [54]. In diabetes mellitus, Akt2 is downregulated. Moreover, these researchers reported that insulin stimulates Akt2 binding ATP7A, which next activates SOD3 by Cu [55]. Both the increased secretion of SOD3 and the translocation of the ATP7A transporter to the plasma membrane of the arterial endothelial cell will ultimately allow the active SOD3 to neutralize the O_2_-● radical allowing NO to act on the arterial endothelium, finally relaxing it [54,55]. Therefore, this prevents the problems of atherosclerosis and cardiovascular disease, in general, associated with DM. Moreover, in the present study, as we have pointed out previously, an increase in the muscle level of Zn, whose insulin-mimetic action is mediated by its transporters such as Zip (1–6, 8, 10, and 14) [14] located in the plasma membrane or Zip7 mainly located in the membrane of the endoplasmic reticulum and the Golgi apparatus occurred [14]. Therefore, a possible explanation, which would have to be checked in future studies, is that the increase at the cytoplasmic level of Zn in the muscle cell could activate Akt2. Therefore, the whole mechanism of action referred above of enhancing and restoring Akt2-ATP7A-SOD3 pathways, could be related to the present findings of increased muscle Cu and Zn levels after melatonin supplementation in this animal model of obesity and T2DM.

Nevertheless, the design of this study has two main limitations: First, we did not measure insulin resistance. However, in a previous study, we measured insulin resistance and glucose homeostasis in the same animal model of diabesity [10]. Second, the hypotheses for SOD1 have not been measured in the analyzed tissues. Although, we previously measured this enzyme activity in renal tissue [20], finding that melatonin-supplemented rats showed greater SOD activity in renal mitochondria of lean and diabetic obese ones. On the other hand, in the study design, we used a daily dose of melatonin of 10 mg/kg body weight. These limitations as well as a possible more appropriate melatonin dose should be considered in the design of future studies on this subject.

## 5. Conclusions

This is the first study to demonstrate that chronic melatonin supplementation in the animal model of obesity and the associated T2DM represented by ZDF rats increased Zn and Cu levels in skeletal muscle cells. This, at least partially, and in concert with our previously reported antihyperglycemic and antihyperinsulinemic effects in the same strain animal model, support that melatonin supplementation improves conditions of diabetes associated to obesity. Further in vitro and/or in vivo studies are required to fully understand the possible effect of melatonin through Zn and Cu levels overall in skeletal muscle in obesity associated diabetes, as well as decipher the precise underlying related molecular mechanism.

## Figures and Tables

**Figure 1 pharmaceutics-13-01535-f001:**
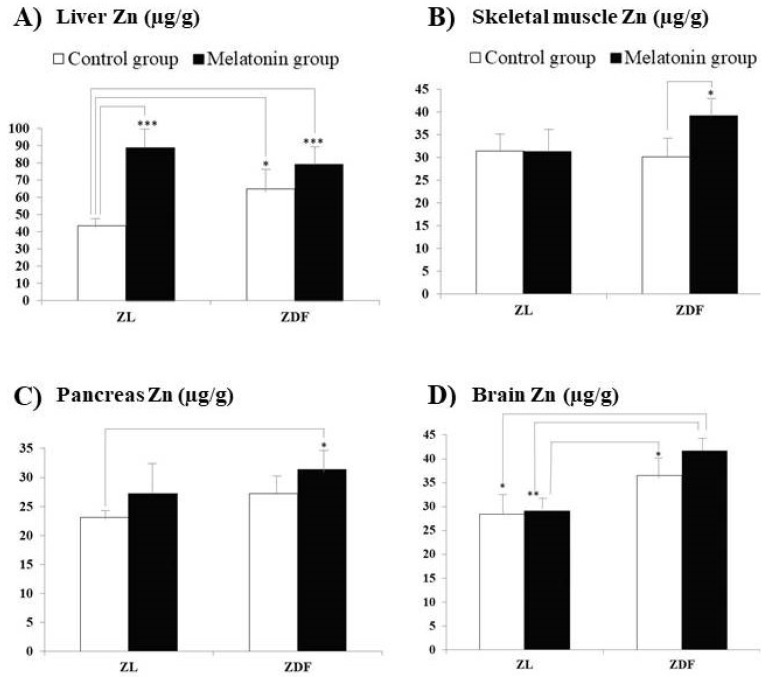
Influence of melatonin supplementation in Zn levels of different organ tissues in male Zücker lean and diabetic fatty rats. (**A**) Liver Zn. (**B**) Skeletal muscle Zn. (**C**) Pancreas Zn. (**D**) Brain Zn. C–ZL: Control lean rats without melatonin; M–ZL: Lean rats with melatonin; C–ZDF: Control diabetic fatty rats without melatonin; M–ZDF: Diabetic fatty rats with melatonin; ZL: Zücker lean rats; ZDF: Zücker diabetic fatty rats. Values are means ± SEM (*n* = 10) of ratios of Zn levels. * *p* < 0.05; ** *p* < 0.01; *** *p* < 0.001 (Tukey post hoc test).

**Figure 2 pharmaceutics-13-01535-f002:**
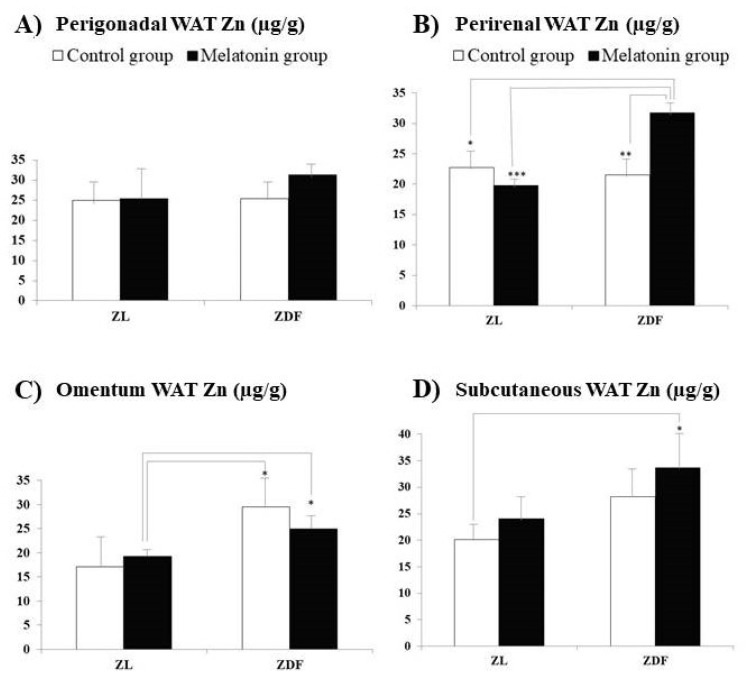
Influence of melatonin supplementation in Zn levels of white adipose tissues (WAT) in male Zücker lean and diabetic fatty rats. (**A**) Gonadal WAT Zn. (**B**) Renal WAT Zn. (**C**) Omentum WAT Zn. (**D**) Subcutaneous Zn. C–ZL: Control lean rats without melatonin; M–ZL: Lean rats with melatonin; C–ZDF: Control diabetic fatty rats without melatonin; M–ZDF: Diabetic fatty rats with melatonin; ZL: Zücker lean rats; ZDF: Zücker diabetic fatty rats. Values are means ± SEM (*n* = 10) of ratios of Zn levels. * *p* < 0.05; ** *p* < 0.01; *** *p* < 0.001 (Tukey post hoc test).

**Figure 3 pharmaceutics-13-01535-f003:**
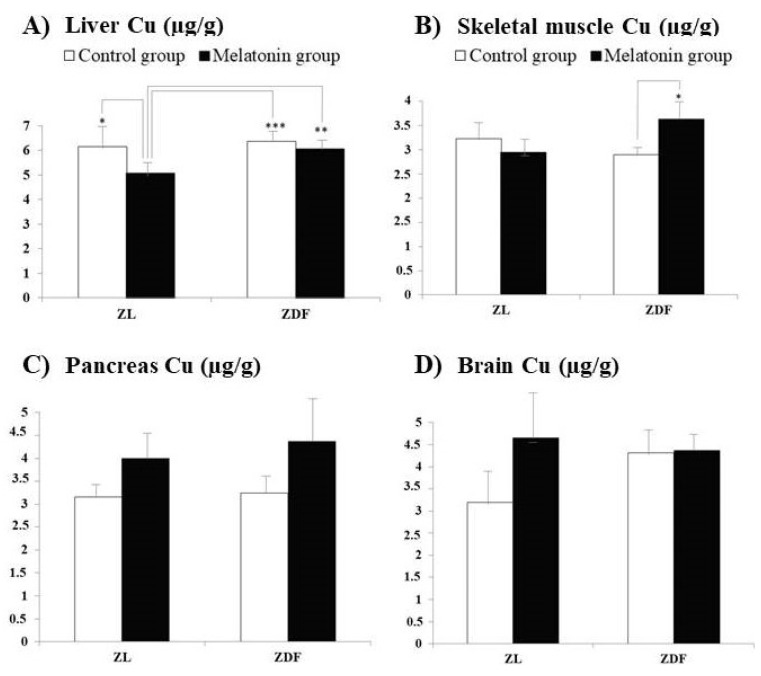
Influence of melatonin supplementation in Cu levels of different organ tissues in male Zücker lean and diabetic fatty rats. (**A**) Liver Cu. (**B**) Skeletal muscle Cu. (**C**) Pancreas Cu. (**D**) Brain Cu. C–ZL: Control lean rats without melatonin; M–ZL: Lean rats with melatonin; C–ZDF: Control diabetic fatty rats without melatonin; M–ZDF: Diabetic fatty rats with melatonin; ZL: Zücker lean rats; ZDF: Zücker diabetic fatty rats. Values are means ± SEM (*n* = 10) of ratios of Zn levels. * *p* < 0.05; ** *p* < 0.01; *** *p* < 0.001 (Tukey post hoc test).

**Figure 4 pharmaceutics-13-01535-f004:**
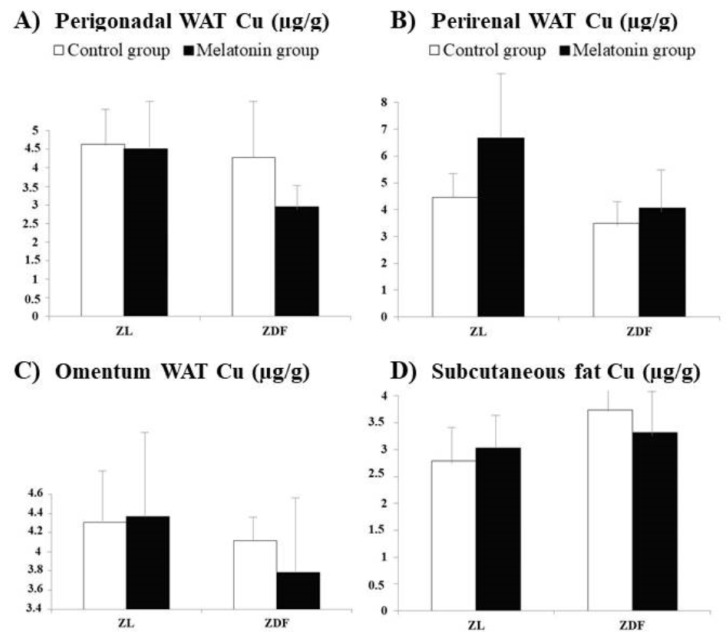
Influence of melatonin supplementation in Cu levels of different white adipose tissues (WAT) in male Zücker lean and diabetic fatty rats. (**A**) Gonadal WAT Cu. (**B**) Renal WAT Cu. (**C**) Omentum WAT Cu. (**D**) Subcutaneous Cun. C–ZL: Control lean rats without melatonin; M–ZL: Lean rats with melatonin; C–ZDF: Control diabetic fatty rats without melatonin; M–ZDF: Diabetic fatty rats with melatonin; ZL: Zücker lean rats; ZDF: Zücker diabetic fatty rats. Values are means ± SEM (*n* = 10) of ratios of Zn levels (*p* > 0.05; Tukey post hoc test).

## Data Availability

Data is contained within the article.

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
