# Peer review of "Melatonin Improves Levels of Zn and Cu in the Muscle of Diabetic Obese Rats"

_pharmaceutics, 2021, doi:10.3390/pharmaceutics13101535_

Round 1

Reviewer 1 Report

It is an interesting study on the influence of melatonin on Zn and Cu levels in obese diabetic rats. Only some considerations:

-Why is the dose of melatonin of 10mg / kg used? Could there be changes with another dose?. You say: and the melatonin dose was adjusted for body weight over the entire period of the study but this was not reflected later if it had to be modified.

-The results become difficult to read, it is preferable that they talk about Zn and put the graphs and after Cu and put the graphs

-In the figures it would be an added value to put asterisks when it is significant and different asterisks according to the degree of significance

-The discussion is very interesting because it comments on the positive and negative results well, but the limitations of the study, its design, and whether this could have influenced should be reflected.

-

Author Response

Manuscript ID: pharmaceutics-1374956

Manuscript title: Melatonin improves levels of Zn and Cu in the muscle of diabetic obese rats

First of all, we would like to thank the reviewer’ comments and suggestions because they have considerably improved and clarified some important aspects of the manuscript. All changes done along the manuscript have been highlighted in yellow color as you can check in its revised version.

AUTHORS REPLY TO THE REVIEW REPORT (REVIEWER 1)

  • Fine/minor spell check for English language and style have been done in the revised version of the manuscript.
  • We really thank reviewer’s statements on “It is an interesting study on the…”.

  • Why is the dose of melatonin of 10mg / kg used? Could there be changes with another dose?. You say: and the melatonin dose was adjusted for body weight over the entire period of the study but this was not reflected later if it had to be modified.

* Along the several studies performed by our research group during the last 12 years on the beneficial effects on melatonin supplementation on the physiological status of this animal model of diabesity (ZDF rats) we have always used the daily dose of 10 mg/kg body weight. This was due because when we began the research in the melatonin field we checked that this dose was the best for the weight loss and the regulation of the thermogenesis by enhancing the UCP expression in the inguinal white adipose tissue in the Zucker diabetic fatty rats.

* We do not know if it could there are any changes with another melatonin dose.

* Following reviewer’s comments we have changed in the revised version the statements on “melatonin dose was adjusted for body weight….(page 4, lines 146-147 in the original version). It has been changed to:

  • Page 3, lines 125-126 (revised version): “and the melatonin dose was adjusted for body weight over the entire period of the study [37], taking into consideration that every two days the rats were weighted again and corresponding adjustments were made…

  • The results become difficult to read, it is preferable that they talk about Zn and put the graphs and after Cu and put the graphs

* Results’ section: Following reviewer’s suggestions we have talk on Zn and put their graphs first, and after we have done the same for Cu.

  • In the figures it would be an added value to put asterisks when it is significant and different asterisks according to the degree of significance

* Figures: As the reviewer recommended we have added values for the asterisks (located in the figure legends as well as in the Statistical Analysis’ section) when it was significant and different asterisks depending on the degree of significance.

  • The discussion is very interesting because it comments on the positive and negative results well, but the limitations of the study, its design, and whether this could have influenced should be reflected.

* Following the reviewer’s statements the main limitations of the study design has been added at the end of the discussion section (pages 11, lines 416-424; revised version) namely:

  • Nevertheless, the design of this study has two main limitations: first we did not measure insulin resistance; nevertheless, in a previous study we measured insulin resistance and glucose homeostasis in the same animal model of diabesity [10]; second, hypotheses for SOD1 have not either measured in analyzed tissues, although we previously measured this enzyme activity in renal tissue [20], finding that melatonin-supplemented rats showed greater SOD activity in renal mitochondria of lean and diabetic obese ones. On the other hand in the study design we used a daily dose of melatonin of 10 mg/kg body weight. These limitations as well as a possible more appropriate melatonin dose should be considered in the design of future studies on this subject.

Sincerely yours,

Miguel Navarro-Alarcón, Professor Dr.Department of Nutrition and Food Science Faculty of Pharmacy University of Granada, GranadaSpain

Reviewer 2 Report

Current study evaluated melatonin supplementary can increase Zn, Cu levels in rats.

This animal study showed that PO melatonin supplement increased Zn and Cu levels, especially in diabetic fatty rats.

Overall, it is rated as a well-done study. However, I would like to comment on a few points in the manuscript.

1) Introduction:

There is too narrative and much contents. Large portion of introduction seems to be more suitable in Discussion section.

2) Method, statistical analysis

Authors acknowledged that Variance analysis, Duncan' multiple range tests, ANOVA, Mann-Whitney test were used. When I see the result data, analysis was done with ANOVA and post-hoc Turkey test. If the statistical technique in the method was actually used, please additionally specify where the comparison was used. In addition, provide overall p-value in ANOVA test.

3) Discussion

Current study evaluated the level of Cu and Zn according to melatonin supplement. Discussion contained large portion of how Cu and Zn may influence insulin resistance and metabolism. However, current study finding was should be more focused melatonin and the level of Cu and Zn. In addition, discussion should be more shorten, this is not a review paper.

Although it has been explained that melatonin can affect insulin metabolism by affecting Cu and Zn based on the results of other studies, this study itself is not a study that measures insulin resistance. Possible hypotheses such as SOD1 have been suggested, but they have not been measured. We recommend that you clearly state these limitations.

Author Response

First of all, we would like to thank the reviewer’ comments and suggestions because they have considerably improved and clarified some important aspects of the manuscript. All changes done along the manuscript have been highlighted in yellow color as you can check in its revised version.

AUTHORS REPLY TO THE REVIEW REPORT (REVIEWER 2)

  • We again really thank reviewer’s statements on “Overall, it is rated as a well-done study. However…”

  • 1) Introduction:

There is too narrative and much contents. Large portion of introduction seems to be more suitable in Discussion section.

* Introduction: Following the reviewer’s statements a large portion on the introduction’s section has been erased in the revised version as well as several references (the original manuscript had 88 and the revised one 61). Specifically 20 lines have been erased in the introduction section (revised version vs. original version).

  • 2) Method, statistical analysis

Authors acknowledged that Variance analysis, Duncan' multiple range tests, ANOVA, Mann-Whitney test were used. When I see the result data, analysis was done with ANOVA and post-hoc Turkey test. If the statistical technique in the method was actually used, please additionally specify where the comparison was used. In addition, provide overall p-value in ANOVA test.

* Method, statistical analysis: Following the reviewer’s statements corresponding changes have been made in the revised version of the manuscript as you can check.

  • Statistical analysis (page 4 lines 153-155; revised version): “Variance analysis and post hoc Tukey test were used for the comparison among groups. The p < 0.05 was statistically significant and levels of significance were labeled on the figure legends as follows: *p < 0.05; **p < 0.01; ***p < 0.001.
  • Result’s section and figures: Additionally specific p values found with the ANOVA test have been included in the text of this section as well as in the figure legends.

  • 3) Discussion

Current study evaluated the level of Cu and Zn according to melatonin supplement. Discussion contained large portion of how Cu and Zn may influence insulin resistance and metabolism. However, current study finding was should be more focused melatonin and the level of Cu and Zn. In addition, discussion should be more shorten, this is not a review paper.

* Discussion: As the reviewer recommended we have erased in the revised version a large portion of comments on how Zn and Cu may influence insulin resistance and metabolism focusing it more on the melatonin and the level of Cu and Zn. In fact 54 lines have been erased in the revised version of the manuscript in the discussion’s section as well as corresponding references (from 88 to only 61 in the revised version) such as you can check in this revised version of the manuscript.

Although it has been explained that melatonin can affect insulin metabolism by affecting Cu and Zn based on the results of other studies, this study itself is not a study that measures insulin resistance. Possible hypotheses such as SOD1 have been suggested, but they have not been measured. We recommend that you clearly state these limitations.

* Following the reviewer’s statements the main limitations of the study design have been added at the end of the discussion section (pages 11, lines 416-424; revised version) namely:

  • Nevertheless, the design of this study has two main limitations: first we did not measure insulin resistance; nevertheless, in a previous study we measured insulin resistance and glucose homeostasis in the same animal model of diabesity [10]; second, hypotheses for SOD1 have not either measured in analyzed tissues, although we previously measured this enzyme activity in renal tissue [20], finding that melatonin-supplemented rats showed greater SOD activity in renal mitochondria of lean and diabetic obese ones. On the other hand in the study design we used a daily dose of melatonin of 10 mg/kg body weight. These limitations as well as a possible more appropriate melatonin dose should be considered in the design of future studies on this subject.

Sincerely yours,

Miguel Navarro-Alarcón, Professor Dr.Department of Nutrition and Food Science Faculty of Pharmacy University of Granada, GranadaSpain

Round 2

Reviewer 2 Report

Thank you for your sincere editing efforts.